# Novel three-dimensional biochip pulmonary sarcoidosis model

Tess M. Calcagno[1], Chongxu Zhang[2], Runxia Tian[2], Babak Ebrahimi[3], Mehdi Mirsaeidi[2]*

1 Department of Medicine, University of Miami, Miami, FL, United States of America, 2 Division of Pulmonary and Critical Care, University of Miami, Miami, FL, United States of America, 3 Research and Development, Genix-Engineering, Irvin, California, United States of America

* msm249@med.miami.edu

## Abstract

Sarcoidosis is a multi-system disorder of granulomatous inflammation which most commonly affects the lungs. Its etiology and pathogenesis are not well defined in part due to the lack of reliable modeling. Here, we present the development of an *in vitro* three-dimensional lung-on-chip biochip designed to mimic granuloma formation. A lung on chip fluidic macrodevice was developed and added to our previously developed a lung-on-membrane model (LOMM). Granulomas were cultured from blood samples of patients with sarcoidosis and then inserted in the air-lung-interface of the microchip to create a three-dimensional biochip pulmonary sarcoidosis model (3D BSGM). Cytokines were measured after 48 hours. ELISA testing was performed to measure cytokine response difference between LOMM with 3D BSGM. There were statistically significant differences in IL-1ß (P = 0.001953), IL-6 (P = 0.001953), GM-CSF (P = 0.001953), and INF-γ expressions (P = 0.09375) between two groups. The current model represents the first 3D biochip sarcoidosis model created by adding a microfluidics system to a dual-chambered lung on membrane model and introducing developed sarcoid-granuloma to its air-lung-interface.

**Data Availability Statement:** All relevant data are within the manuscript and its Supporting Information files.

**Funding:** This work was supported in part by a Health Resources and Services Administration

## Introduction

Sarcoidosis is a heterogenous multisystem disorder characterized by the formation of noncaseating granulomas which can lead to extensive fibrosis. Its etiology is still largely unknown, though genetic, environmental, and infectious causes have all been considered [1]. Its clinical presentation is extremely variable ranging from an asymptomatic to a life-threatening disease [2]. Although sarcoidosis can virtually affect any organ, lung is the most involved organ and is associated with the highest morbidity and mortality, usually related to and extensive pulmonary fibrosis and pulmonary hypertension [3].

The features of human pulmonary sarcoidosis that could be modeled are still undefined, largely due to the limited understanding of etiology and pathogenesis of the disease. Human exposure to a wide variety of infectious and environmental components including *mycobacteria* [4–8], *propionibcateria* [9,10], occupational antigens [11], desert dusts [12,13], and also to human cancer cell products [14] may cause sarcoidosis with different clinical phenotypes [15].

contract awarded to MM (234–2005–37011C), and Mallinckrodt Pharmaceuticals for research grant support awarded to MM (Grant No. 5043). Genix-Engineering provided support for this study in the form of a salary for BE. The specific roles of this author are articulated in the 'author contributions' section. The content is the responsibility of the authors alone and does not necessarily reflect the views or policies of the Department of Health and Human Services, nor does mention of trade names, commercial products, or organizations imply endorsement by the U.S. Government. The funders had no role in study design, data collection and analysis, decision to publish, or preparation of the manuscript.

**Competing interests:** The authors have read the journal's policy and the authors of this manuscript have the following competing interests: BE is a paid employee of Genix-Engineering. MM was awarded a research grant from Mallinckrodt Pharmaceuticals and is an advisory board member of Mallinckrodt Pharmaceuticals. MM has a pending patent for this technology (Provisional patent application UMIP-441 Mirsaeidi 126683-040PV1). This does not alter our adherence to PLOS ONE policies on sharing data and materials.

Inherited histocompatibility complexes including HLA-DR allele DRB1*1101 in African American and European American is associated with the development of sarcoidosis [16]. Genetic polymorphisms in Angiotensin-Converting enzyme have also been linked to the development and degree of severity of sarcoidosis [17]. Mutation in *annexin A11* has been associated with susceptibility to sarcoidosis and pulmonary fibrosis in sarcoidosis [18,19]. Genetic predisposition to sarcoidosis is an important consideration in developing an acceptable animal model, which is currently not well developed. Even if animal granulomas are created via the introduction of infectious/environmental agents, such granulomas may represent only a subgroup of sarcoidosis patients.

Two-dimensional *in vitro* cell culture models combined with *in vivo* animal testing served as gold standard methods for decades of medical advancements in pulmonary research [20]. In two-dimensional (2D) cell culture, cells of one type can be studied in a low-cost standardized fashion which allows for massive high throughput screenings and experimental protocols which can easily be replicated. However, 2D cell culture is limited by its fixed device architecture and stagnant culture media. Simplistic representation fails to recreate lung-specific microenvironments which rely on the interplay between multiple cell types within a complex physiologic system [21].

Contemporary three-dimensional (3D) microfluidic lung on chip models more accurately reconstruct normal lung physiology [22,23]. Lung on chip models use microfluid-based cell culture on a micro- or nano-sized chip allowing for perfusion of targeted cells into the chip and reduction of reagent consumption as compared to conventional cell culture technique. Most lung-on-chip models (LOCM) are designed to mimic the environment of the alveolar air-blood barrier comprised of alveolar epithelial cells and vascular endothelial cells in the setting of applied mathematical models which consider both fluid mechanics and stretch frequency [24]. Structural limitations of cell culture are largely circumvented if the cultured cells are introduced to a 3D microfluidics model. For example, cultured primary human cell lines with *M. tuberculosis* have been added to 3D microfluidic plates to model active granulomatous inflammation [25].

In the setting of sarcoidosis, novel 3D models properly mimicking granuloma formation have not yet been created; this has contributed to the lack of available targeted therapies. We developed a novel lung on chip model by adding a microfluidics system to our previously developed dual-chambered lung on membrane model (LOMM) [26,27]. We also recently developed an in vitro sarcoidosis granulomas model [28]. Here, we introduced the in vitro granuloma to the air-lung interface (ALI) of LOMM, making this the first three-dimensional lung on chip model mimicking sarcoidosis.

## Methods

### Development of granuloma

Human peripheral blood mononuclear cells (PMBCs) were introduced to granuloma-inducing microparticles to effectively create human sarcoid-like granulomas. PMBCs were isolated from peripheral blood samples of patients with confirmed sarcoidosis taken from University of Miami Sarcoidosis biobank as previously described [28]. All experiments were performed in accordance with relevant guidelines and regulations and approved by University of Miami Institutional Review Board no 20150612. Informed consent was obtained from all subjects in this study. A quantity of $2 \times 10^6$ PBMC was cultured in a 12-well tissue culture plate and immediately challenged with microparticles generated from *myobacterium abscessus (MAB)* cell wall to induce granuloma formation. The methods for microparticle development and details of

granuloma formation were discussed elsewhere [28]. Granuloma maturation was considered complete when the diameter of the based was in the range of 100 to 200 micrometers.

### Design and fabrication of fluidic macrodevice

The lung on chip fluidic macrodevice is comprised of two main components, the base and top cover (cap). Both components are CNC machined Polycarbonate blocks that sealed together to create a multi-channel platform with independent flow-control capability. There are three channels at the base, where cell cultures are inserted via membrane dual cell culture plate (12 mm Transwell with 0.4 μm pore polycarbonate membrane insert). Channels are independently sealed via Soft Viton® Fluoroelastomer O-Ring (Made of FDA Listed Material, SAE AS568). For each channel at the base, there is a corresponding channel in the top cover, providing controlled air supply to the cell culture. Ismatec REGLO ICC peristaltic pump (3 channel/3 roller) was utilized to independently control the flow rate of the 3 channels for media flow and 3 channels for air flow. If needed, a separate pump could be used to achieve independent air flow control. Masterflex Tygon E-LFL tubes with 1.42mm ID were used for fluid transfer. Considering 1ml medium volume in each chamber and 700mm tube length per channel, the pump speed was set to 20 rpm to achieve 2ml/min flow rate per channel resembling lung cell exposure to blood based on a cardiac output equal 5–6 liter/min. 6 electrodes (3 in the top cover and 3 in the base) were used for continuous electrical impedance measurement (TEER test). The microfluidics system allows the researcher to control the flow rate in real time using an iOS and Android compatible microcontroller.

### Development of lung on chip

We previously developed a lung-on-membrane model (LOMM) with a dual chamber including normal bronchial epithelial (NHBE) cells and human microvascular endothelial cells [26,27]. We added a microfluidics system to the LOMM to develop a novel lung on chip.

### Adding granuloma to ALI

Fully matured granulomas were developed separately. We then added granuloma to the ALI of the chip included with a lung-on membrane model. A narrow scratch was made on the middle of the ALI surface of membrane and developed granulomas from $2 \times 10^6$ PBMC in 50 μL of medium were directly added to the narrow to develop three-Dimensional Biochip Sarcoidosis Granuloma Model (3D BSGM).

### Cytokine measurements

To demonstrate the functionality of the device to run an experimental study, we developed ten exact replicate LOMM and ten exact replicate 3D BSGM as previously described. The circulatory medium of each unit was collected after 48 hours and ELISA testing was performed to measure cytokine response difference between groups. ELISA was performed using a kit from Invitrogen (Carlsbad, CA, USA) (Procartaplex human th1/th2 cytokine panel 11 plex from Invitrogen, cat # epx110-10810-901) per the manufacturers' instruction. We designed the experiment with 3 replicates, each replicate has 10 samples.

## Results

Mature granulomas from PBMC from sarcoidosis patients and challenged with MAB microparticles developed within 72 hours as shown in Fig 1. The presence of multi-nucleated giant

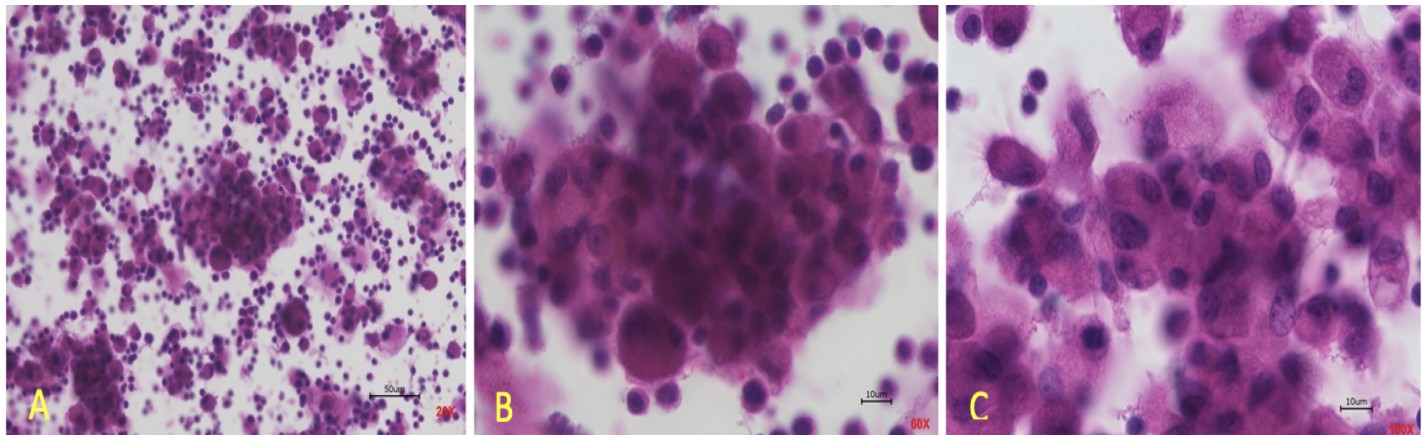

**Fig 1. *In vitro* granulomas from PBMC taken from blood of patients with sarcoidosis.** (A) displays granuloma at 20X magnification, (B) displays granuloma at 60X magnification, (C) displays granuloma at 100X magnification.

cells, lymphocytes, and macrophages which are aggregated together formed a large- structured granuloma.

The fluidic macrodevice was developed and shown in Fig 2. S1 Fig shows TEER test functionality of the device.

The bilayer lung model was developed as shown in Fig 3 which displays TEM images of the lung model and its varying components (NHBE cells, endothelial cells, and intervening membrane). We also observed small vesicles (sized about 300 μm, assumed exosomes) into channels between epithelial and endothelial cell layers as shown in Fig 3C.

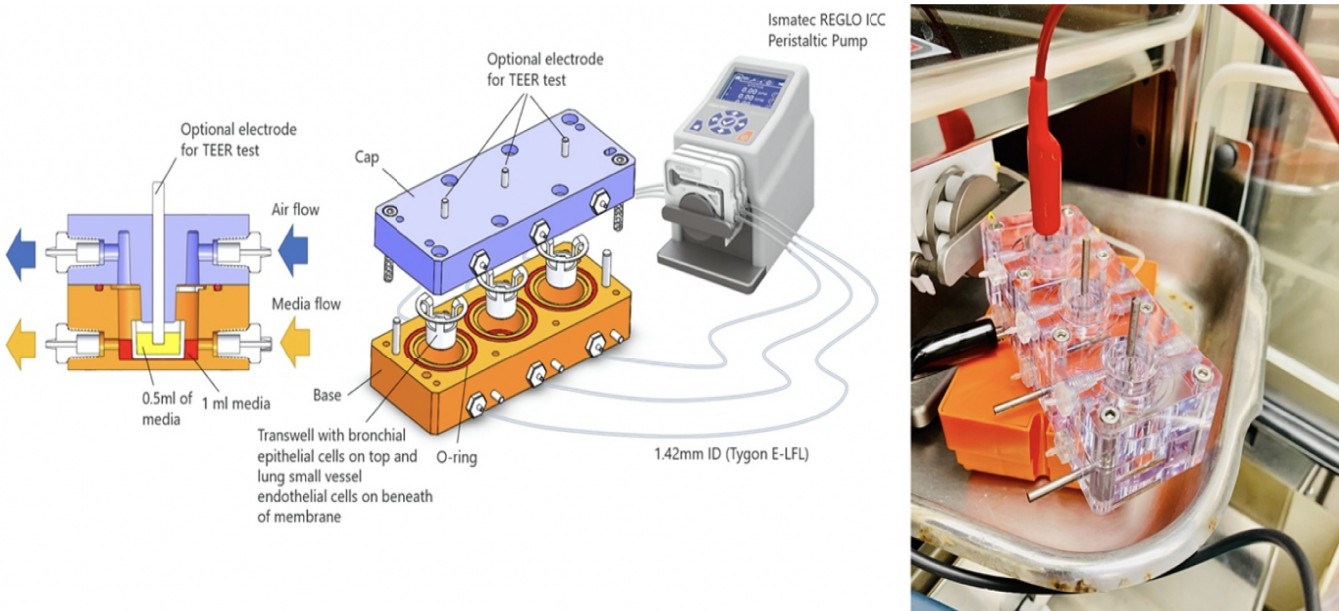

**Fig 2. Design and fabrication of fluidic macrodevice and physical device in action.** The macrodevice set up in its compact form contains three individually controlled channels which are independently sealed. Each channel is equipped with integrated electrodes for resistance measurement, in-line flow meters, microcontrollers, and built-in Wi-Fi connectors.

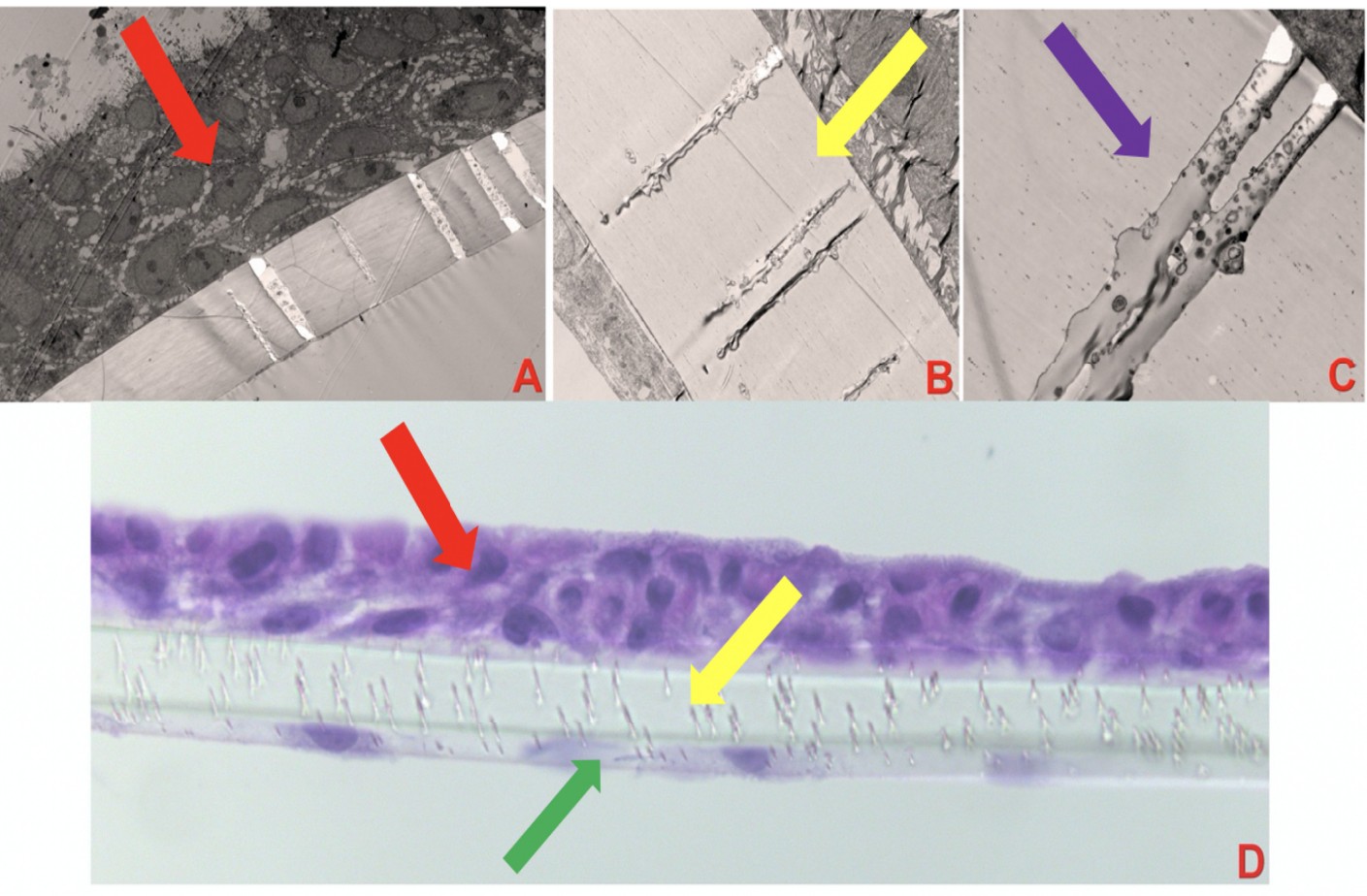

**Fig 3. Dual-chambered lung on membrane model (LOMM).** (A) shows LOMM with the red arrow pointing to NHBE cells, magnification X400, exposure 3000 (ms). (B) shows LOMM with the yellow arrow pointing to the polycarbonate membrane magnification X1500, exposure 3000 (ms). (C) shows LOMM with the purple arrow pointing to membrane with *0.4* μm pore magnification X4000, Exposure 3000 (ms). (D) shows LOMM displaying NHBE cells (red arrow), membrane (yellow arrow), and endothelial cells (green arrow), magnification X40.

The three-dimensional biochip pulmonary sarcoidosis model (BOSGM) developed from the addition of preformed granulomas to the bilayer lung model was shown in Fig 4.

To test the functionality of the BOSGM, cytokine measurement was performed in media collected from the BSGM and compared with control (lung on chip without granuloma). There were statistically significant differences in IL-1ß expression (P = 0.001953), IL-6 expression (P = 0.001953), GM-CSF expression (P = 0.001953), and INF-γ expression (P = 0.09375) as shown in Fig 5.

## Discussion

We developed a Three-Dimensional Biochip Sarcoidosis Granuloma Model by adding a microfluidics system to our previously developed lung on membrane model and introduced fully developed granulomas to its air-lung interface. Our model uses standardization techniques and is compatible to use artificial intelligence (AI) to combine the benefits of reproducibility from 2D cell culture with the complexity of lung on chip microfluid models. It is comprised of three individually controlled channels which are independently sealed and contain individual peristaltic pumps to control the flow rate of media in each channel with precision.

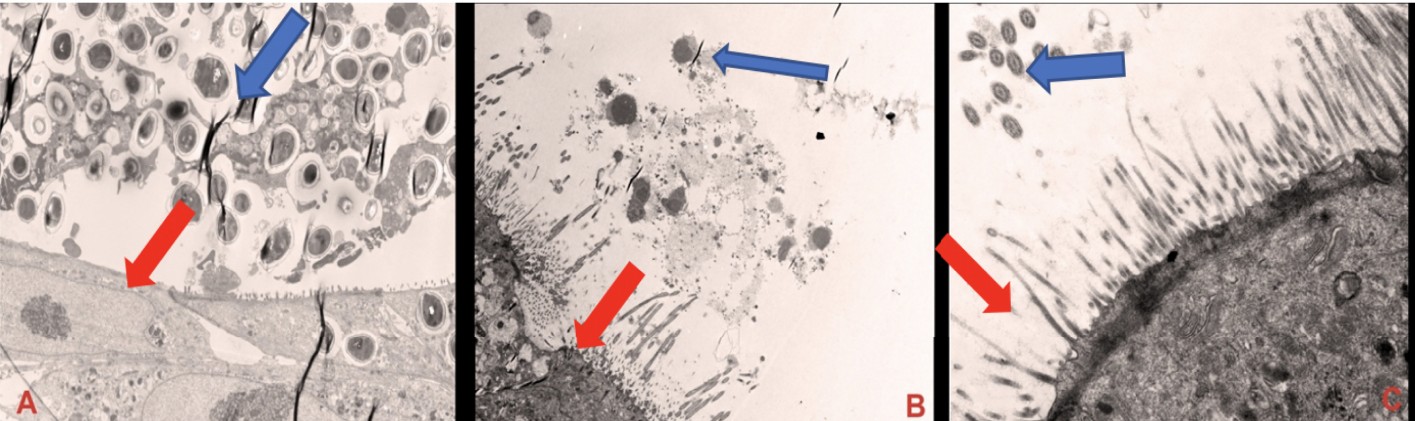

**Fig 4. Transmission electron microscopic images of the three-dimensional biochip pulmonary sarcoidosis model.** (A) shows bronchial epithelial cell with cilia (Red arrow) and granuloma (lymphocytes and macrophages) with Blue arrow, magnification of X1200. (B) shows macrophages and lymphocytes from developed granuloma in the ALI, magnification of x1200, Exposure 3000 (ms). (C) shows mature bronchial epithelial cell with cilia (Blue arrow) and microvillia (Red arrow), magnification X5000. Red arrows show NHBE cells.

Lung on chip (LOC) models emulate lung physiology using a microfluid design combined with culture inoculation [29,30]. Huh *et al.* developed the first LOC model which contains two microchannels coated with endothelial cells and alveolar epithelial cells separated by a 10um collagen coated polydimethylsiloxane (PDMS) porous membrane. Two separate channels with vacuum applied suction surround the lateral aspects of the model to stimulate negative intra-pleural pressure involved in inspiration [31]. Humayun *et al.* added a hydrogel micro layer which allowed for the development of a LOC model with smooth muscle and epithelial cells to mimic bronchiolar microenvironments [32]. This model was limited with being a solid system (unable to remove the membrane) with unscalable experiment due to very limited cell numbers in the system. LOC models have since been used to mimic lung injury, lung inflammation, pulmonary fibrosis, and lung cancer, but not sarcoidosis [33–36]. Lately, microscale infectious granuloma models combining cell culture and microchip technology have recently been reported. Berry *et al.* introduced human immune cells combined with virulent mycobacterial strains to a suspended microfluidics platform to study tuberculosis infection [37]. Bielecka *et al.* developed a microsphere granuloma model equipped with a microfluidic plate to model pharmacokinetics and antimicrobial resistance [25]. Walter *et al.* recently developed an in vitro model to study tuberculous mycobacterial granuloma in central nervous system [38].

The inability to recreate sarcoid granulomas *in animal models*, has severely limited our ability to discover novel pharmacologic therapies, characterize its etiology, and understand its heterogenous clinical presentation. Prior to the development of our model, there were no 3D lung models designed to mimic sarcoid-like granulomatous disease. The pathogenesis of sarcoidosis is associated with CD4[+] T Cell activation, and secretion of cytokines including interferon-γ, tumor necrosis factor (TNF)-α, and transforming growth factor-β [39].

In previous investigations, sarcoid has been studied using biopsies, animal models, and single cell lines. Biopsies of granulomas from diseased patients have been used to characterize sarcoid histologically at a stagnant point in time, but they cannot properly represent the dynamic interplay between multiple cell types seen in sarcoid-like granulomas [40]. Many animal models made to simulate granuloma formation have been developed in the past ten years, but development is limited since sarcoidosis does not occur naturally in most animals. In order to simulate sarcoid-like granulomas in animals, animals are exposed to environmental antigens [15] and mycobacterial antigens [26,41]. For example, Werner *et al.* created a pulmonary

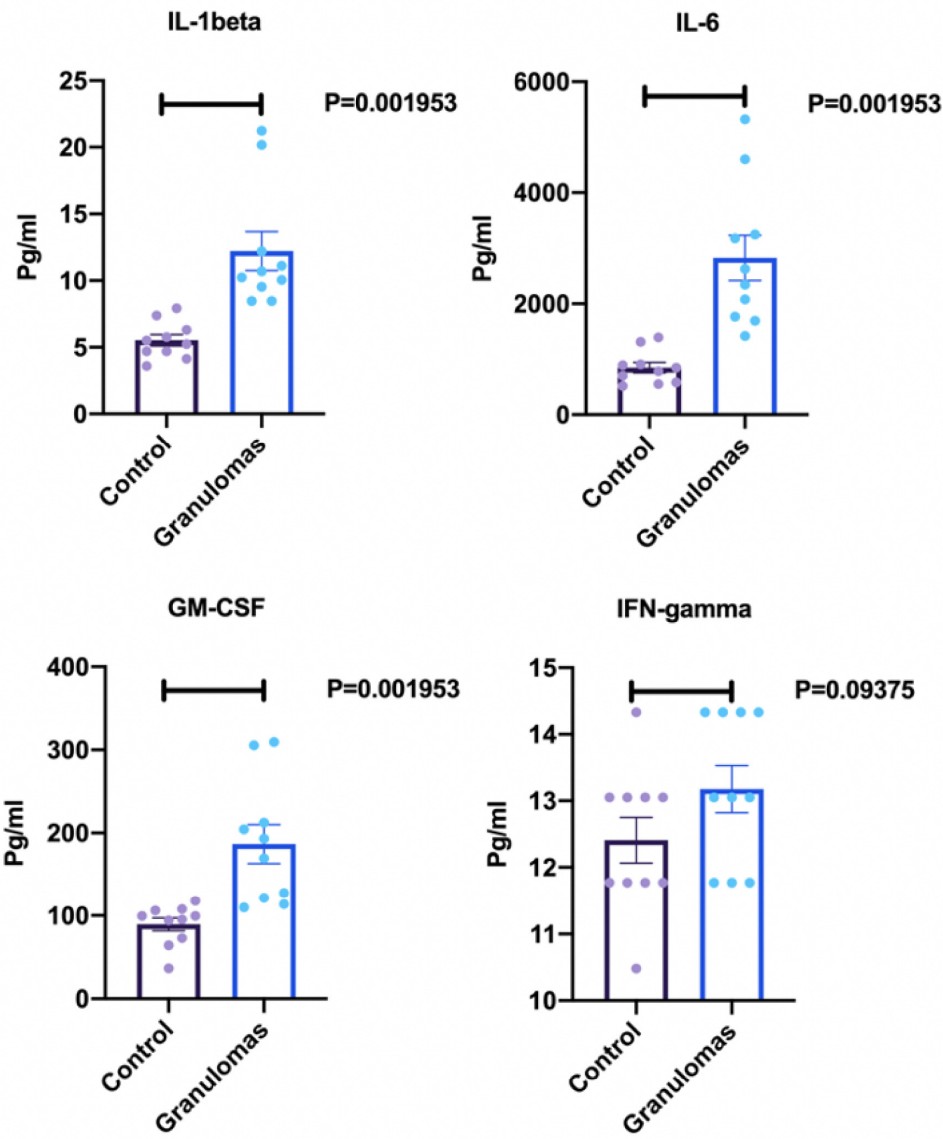

**Fig 5. ELISA results for cytokine response measurements.** Concentrations of cytokines (IL-1ß, Il-6, GM-CSF, and IFN-γ) reported in in pg/ml. Control represents the lung on chip model without granuloma in ALI. Granulomas represent lung on chip model with granulomas in ALI (n = 10 for each group, repeated X3).

granuloma in mice by introducing *Propionibacterium acnes*, an environmental bacterium possibly implicated in the development of sarcoidosis in human lung tissue [42]. Such models have been limited by lack of similarity to polygenic human sarcoid-induced granulomatous processes involving the interplay between T-cell, cytokine, and aberrant macrophage-mediated immune responses [43]. Two-dimensional cell cultures using BAL or blood samples have been used to study antigenic response in a single cell population, however this fails to account for a sarcoid microenvironment incorporating multiple cell types [40].

The current model is designed based on a mycobacteria component which is one of the potential etiologies of sarcoidosis. However, it does not consider granuloma formation in

response to other potential environmental exposures. This may limit future study to one pathophysiologic subset of sarcoid disease.

## Future directions

The proposed model which uses three-dimensional structuring, compatible to AI, and standardization techniques mimics pulmonary sarcoidosis. AI will be the next step of development of this device. AI can control TEER testing, volume control as well as adding new medication of chemical to the experimental chamber. This development circumvents a large hurdle in the progression of sarcoidosis research. The model is scalable to be incorporated with four cell groups including fibroblast and circulatory macrophages. It will allow for better characterization of a heterogenous and complex multi-organ disease. Furthermore, efficient testing of novel pharmacotherapeutic agents will bring agents into clinical trials more quickly.

## Supporting information

**S1 Fig.**
(TIF)

**S1 Data.**
(DOCX)

## Author Contributions

**Conceptualization:** Chongxu Zhang, Babak Ebrahimi, Mehdi Mirsaeidi.

**Methodology:** Chongxu Zhang, Runxia Tian, Babak Ebrahimi.

**Writing – original draft:** Tess M. Calcagno.

**Writing – review & editing:** Tess M. Calcagno, Mehdi Mirsaeidi.

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
