## [Decision Letter · Decision Letter 0]

21 Oct 2020

PONE-D-20-30175

Development of a novel three-dimensional organoid sarcoidosis granuloma model; created by adding a microfluidics system and developed granulomas to a dual-chambered lung on membrane model

PLOS ONE

Dear Dr. Mirsaeidi,

Thank you for submitting your manuscript to PLOS ONE. After careful consideration, we feel that it has merit but does not fully meet PLOS ONE’s publication criteria as it currently stands. Therefore, we invite you to submit a revised version of the manuscript that addresses all the points raised during the review process.

Two experts have evaluated the manuscript. While the data are promising, the manuscript need to be substantially reorganized and rewritten. Both reviewers found that the references do not reflect the field. Ethical approvals/permissions for the use of human cells need to be added to the Methods section.

We look forward to receiving your revised manuscript.

Kind regards,

Mária A. Deli, M.D., Ph.D.

Academic Editor

PLOS ONE

Journal Requirements:

2. You state "to demonstrate the functionality of the device to run an experimental study, we developed ten LOMM and ten 3D OSGM as previously described." Please clarify whether these were exact replicates, or whether the ten LOMMs and OSGMs differed. If so, how did they differ from one another?

"The authors have declared that no competing interests exist".

We note that one or more of the authors are employed by a commercial company: 'Genix-Engineering, Irvine, California, USA'.

Reviewers' comments:

Reviewer's Responses to Questions

**Comments to the Author**

1. Is the manuscript technically sound, and do the data support the conclusions?

Reviewer #1: Yes

Reviewer #2: Partly

2. Has the statistical analysis been performed appropriately and rigorously? 

Reviewer #1: Yes

Reviewer #2: N/A

3. Have the authors made all data underlying the findings in their manuscript fully available?

Reviewer #1: Yes

Reviewer #2: Yes

4. Is the manuscript presented in an intelligible fashion and written in standard English?

Reviewer #1: Yes

Reviewer #2: Yes

5. Review Comments to the Author

Reviewer #1: Cacagno et al. developed an in vitro sarcoidosis granuloma model derived from the blood of patients. These 3D cellular structures were then added to a novel microfluidics system using regular Transwell system to model air liquid interface (ALI), and cell-cell interactions, cell aggregate-cell interactions and cytokine release was characterized.

Generally the paper contains very interesting material related to the field, since no good lung granuloma-lung tissue (or other barrier tissue) in vitro model exists. Although non-referenced, but important papers in this research area should be cited such as (PMIDs): 28174307; 32974300; 32716613. This model is new by combining the ALI with the granuloma formation and studying cell-cell interactions. The major problems with the paper are not the findings, but with the presentation of the data.

Introduction is too general, contains repetitions and the references are not always matching the text in a required fashion. Authors claim that neither human, animal or in vitro models are good to model the disease, which is important when one wants to state the limitations of different experimental systems. Here biochip models are claimed as too low throughput and cell culture models as redundant. Still authors try to present a new microfluidic-ALI-cell culture granuloma-model. Although one understands the model, the reasoning behind is not very logical.

Methods section does not contain any ethical statements, although cells were isolated from patients. This has to be added. Discussion contains some methodology about the device, which should be inserted here. 10 paralells mean less than 4 repeats with the system / group?

Results section is extremely redundant. Results have to be presented more in detail, not just as one would describe a figure legend.

Please also double-check all references, for which I also indicated some ideas:

- Ref 2, 3: the text is matching the logic of the paper, but the references are not very well chosen, they are too general. To discuss the limitations of animal models, special sarcoidosis animal models should be mentioned and discussed. Ref. 3. Is a German language paper matching the content of the text, but is also too general and is not available to read to wider audience.

- Ref 4-7. - About biochip modeling and its versatility. These references are coming from 2 groups, where 2 papers is citing heart tissue monitoring. Plenty of papers are to be found about lung models, and in other tissue barrier models, please use a more broad range of citations when referring to this very wide field.

- Sentence before Ref. 9. is almost the same as sentence before ref.2., so it is a repetition, please remove.

- Refs 12 and 13 are the same, please correct.

In general the paper contains good data, and the biochip system looks valuable, but the manuscript has to be re-formatted, carefully checked and some parts re-written. Also the title (and the text) should be re-formatted: the word "organoid" suggests a stem-cell based approach, which is not the case here.

Reviewer #2: The authors present the development of an in vitro model of 3-D lung-on-chip organoid designed to mimic granuloma formation, which is characteristic of a heterogeneous multisystem disorder known as sarcoidosis. The 3-D organoid sarcoidosis model was created by adding a newly designed fluidic device, with three channels for cell culture insertion, to a previously developed dual-chambered lung-on-membrane model. Granulomas were cultured from blood samples of patients with sarcoidosis and then inserted in the air-lung interface of a microchip to create a 3-D organoid sarcoidosis model, and it was tested for cell viability with fibroblasts. They further challenged the air-lung interface of the organoid sarcoidosis model with microparticles to demonstrate the functionality of the device for experimental studies.

Overall, the manuscript seems to be prepared in haste without sufficient detail for the readers to follow the narrative resulting in major deficiencies and, therefore, is not recommended for publication in its current form. The field of lung-on-a-chip has rapidly been developing in the last decade and is reaching maturity to a certain extent. Introducing granuloma formation to mimic in vitro sarcoidosis in the lungs is certainly a nice contribution. However, the text is rather confusing as to the origin of granuloma formation; is it from PBMC taken from blood of patients with sarcoidosis or is it induced by in-house developed microparticles?

The images in Figure 1 seem to be taken from a culture dish; how these cultures are utilized for the development of 3-D OSGM? No protocol has been detailed.

The 3 chambers in the system design shown in Figure 2 is an artefact as all chambers are identical; it is rather easy to design a similar device with 6 chambers or assemble two devices with the same number of devices. The authors should completely mute this aspect of the device.

How the cross-section in Figure 3 was taken; presumably this is taken from the previously developed dual-chambered lung-on-membrane model which is not new?

The endothelial layer seems to be rather sparse not quite fully confluent; is this an important issue, Can this be a reliable model?

What is the role of the fibroblast cells shown in Figure 4; the lung model has already been developed and includes only epithelial and endothelial cells. The fibroblast discussion is completely superficial and should be removed although the image is nice. A similar image for the endothelial cells would be much more desired.

6. PLOS authors have the option to publish the peer review history of their article (what does this mean?). If published, this will include your full peer review and any attached files.

Reviewer #1: No

Reviewer #2: No

---

## [Author Response · Author response to Decision Letter 0]

2 Nov 2020

October 23, 2020

Dr. Mária A. Deli, M.D., Ph.D.

 Associate Editor

Plos one

Dear editor,

Thank you for your careful review of our manuscript, and especially for the constructive criticism of the reviewers. We are also grateful for the opportunity to revise the manuscript, and respond to their critiques. We believe that we have addressed all concerns and, in doing so, have significantly improved the manuscript. We understand that our manuscript is not perfect, but, has important messages for your readers. A point-by-point response to items raised by the reviewers follows.

Response to Editors.

Response to Reviewer

Reviewer #1: 

Cacagno et al. developed an in vitro sarcoidosis granuloma model derived from the blood of patients. These 3D cellular structures were then added to a novel microfluidics system using regular Transwell system to model air liquid interface (ALI), and cell-cell interactions, cell aggregate-cell interactions and cytokine release was characterized. Generally the paper contains very interesting material related to the field, since no good lung granuloma-lung tissue (or other barrier tissue) in vitro model exists. 

Although non-referenced, but important papers in this research area should be cited such as (PMIDs): 28174307; 32974300; 32716613. 

Reply: Thanks for reference! It has been added.

The major problems with the paper are not the findings, but with the presentation of the data.

Reply: Thanks for comment. We restructured the manuscript. 

 Introduction is too general, contains repetitions and the references are not always matching the text in a required fashion. Authors claim that neither human, animal or in vitro models are good to model the disease, which is important when one wants to state the limitations of different experimental systems. Here biochip models are claimed as too low throughput and cell culture models as redundant. Still authors try to present a new microfluidic-ALI-cell culture granuloma-model. Although one understands the model, the reasoning behind is not very logical.  

Reply: Thanks for comment. We manuscript has been revised per your comment. 

Methods section does not contain any ethical statements, although cells were isolated from patients. This has to be added. 

Reply: Thanks for comment. Ethical statement was added. 

Discussion contains some methodology about the device, which should be inserted here. 10 paralells mean less than 4 repeats with the system / group? 

Reply: Thanks for comment. (N) is 10 and has been corrected.

 Results section is extremely redundant. Results have to be presented more in detail, not just as one would describe a figure legend. 

Reply: Thanks for comment and apology for mistakes. The discussion has been rewritten.

Please also double-check all references, for which I also indicated some ideas: - Ref 2, 3: the text is matching the logic of the paper, but the references are not very well chosen, they are too general.

Reply: Thanks for comment. The reference section has been corrected 

 To discuss the limitations of animal models, special sarcoidosis animal models should be mentioned and discussed. 

Reply: Thanks for comment. It has been added.

Ref. 3. Is a German language paper matching the content of the text, but is also too general and is not available to read to wider audience. - Ref 4-7. - About biochip modeling and its versatility. These references are coming from 2 groups, where 2 papers is citing heart tissue monitoring. Plenty of papers are to be found about lung models, and in other tissue barrier models, please use a more broad range of citations when referring to this very wide field. - Sentence before Ref. 9. is almost the same as sentence before ref.2., so it is a repetition, please remove. - Refs 12 and 13 are the same, please correct.

Reply: Thanks for comment. We have enriched the reference section.

 In general the paper contains good data, and the biochip system looks valuable, but the manuscript has to be re-formatted, carefully checked and some parts re-written. Also the title (and the text) should be re-formatted: the word "organoid" suggests a stem-cell based approach, which is not the case here.

Reply: Thanks for comment. The paper has been improved with your comments and title has been changed.

Reviewer #2: 

 They further challenged the air-lung interface of the organoid sarcoidosis model with microparticles to demonstrate the functionality of the device for experimental studies.  Overall, the manuscript seems to be prepared in haste without sufficient detail for the readers to follow the narrative resulting in major deficiencies and, therefore, is not recommended for publication in its current form. The field of lung-on-a-chip has rapidly been developing in the last decade and is reaching maturity to a certain extent. Introducing granuloma formation to mimic in vitro sarcoidosis in the lungs is certainly a nice contribution. However, the text is rather confusing as to the origin of granuloma formation; is it from PBMC taken from blood of patients with sarcoidosis or is it induced by in-house developed microparticles?  

Reply: Thanks for comment. We added more details on the development of in vitro granuloma. 

The images in Figure 1 seem to be taken from a culture dish; how these cultures are utilized for the development of 3-D OSGM? No protocol has been detailed. 

Reply: Thanks for comment. The method section has been rewritten to add the methodology of 3D BOSG. 

 The 3 chambers in the system design shown in Figure 2 is an artefact as all chambers are identical; it is rather easy to design a similar device with 6 chambers or assemble two devices with the same number of devices. The authors should completely mute this aspect of the device. 

Reply: We agree with your great point. Your comment has been applied. 

 How the cross-section in Figure 3 was taken; presumably this is taken from the previously developed dual-chambered lung-on-membrane model which is not new? The endothelial layer seems to be rather sparse not quite fully confluent; is this an important issue, Can this be a reliable model? 

Reply: Thanks for comment. This image is from our published model. Endothelial cells are so thin and easily wall off when the paraffin block gets cut. 

 What is the role of the fibroblast cells shown in Figure 4; the lung model has already been developed and includes only epithelial and endothelial cells. The fibroblast discussion is completely superficial and should be removed although the image is nice. A similar image for the endothelial cells would be much more desired.

Reply: Thanks for comment. We moved fibroblasts experiments to supplemental file.

Thank you

---

## [Decision Letter · Decision Letter 1]

16 Nov 2020

PONE-D-20-30175R1

Novel three-dimensional biochip pulmonary sarcoidosis model

PLOS ONE

Dear Dr. Mirsaeidi,

Thank you for submitting your manuscript to PLOS ONE. After careful consideration, we feel that it has merit but does not fully meet PLOS ONE’s publication criteria as it currently stands. Therefore, we invite you to submit a revised version of the manuscript that addresses all the points raised during the review process.

While the manuscript has been improved, many of the original requests remained unanswered. A thorough revision is needed addressing all the original comments, together with a marked-up copy of the manuscript showing all changes in yellow. It should be clearly described, explained and justified if figures were left out or changed.

We look forward to receiving your revised manuscript.

Kind regards,

Mária A. Deli, M.D., Ph.D.

Academic Editor

PLOS ONE

Reviewers' comments:

Reviewer's Responses to Questions

**Comments to the Author**

1. If the authors have adequately addressed your comments raised in a previous round of review and you feel that this manuscript is now acceptable for publication, you may indicate that here to bypass the “Comments to the Author” section, enter your conflict of interest statement in the “Confidential to Editor” section, and submit your "Accept" recommendation.

Reviewer #1: (No Response)

2. Is the manuscript technically sound, and do the data support the conclusions?

Reviewer #1: Partly

3. Has the statistical analysis been performed appropriately and rigorously? 

Reviewer #1: Yes

4. Have the authors made all data underlying the findings in their manuscript fully available?

Reviewer #1: Yes

5. Is the manuscript presented in an intelligible fashion and written in standard English?

Reviewer #1: Yes

6. Review Comments to the Author

Reviewer #1: Calcagno et al. provided R1 in a timely fashion. Although several major issues

have been addressed, I still do not recommend the publication in this form.

This group has high quality and well written papers, such as PMID 32350353,

which shows an example for this paper's aim as for quality. Although PlosOne

is a lower impact factor journal than the PMID 323250353, PlosOne is Q1 in its

field, therefore high quality of writing and data presentation is expected.

Introduction and Discussion were modified and amended as requested. Still some references

are missing, which for example work with Transwell granuloma models, such as PMID 32716613.

References 28 and 29 (previously 12 and 13) are still the same, were not corrected,

please correct.

Ethical statement was added. Clarification about the repeat numbers was added, although

in the text it is still confusing how many repeats were performed. The Figure legend makes

it more understandable.

The general term "organoid" was removed and corrected.

My two most major concerns are the results section and presentation of the data.

Some figures are missing compared to the previous version and the it is not clear

why or what was corrected. Still the description of the data is very redundant,

although a bit improved compared to the previous version.

Also the answers for the reviewers' questions were extremely redundant and

minimalistic, which is very unprecedented and unacceptable. Several questions

of the previous revisons were not answered or are not found well in the text.

Marked-up copy should have been provided to help the work of the reviewers.

In general the paper was improved majorly. Still major revision is needed,

but if authors correct the mistakes outlined, it would be recommended for publication.

7. PLOS authors have the option to publish the peer review history of their article (what does this mean?). If published, this will include your full peer review and any attached files.

Reviewer #1: No

---

## [Author Response · Author response to Decision Letter 1]

8 Dec 2020

Dec 8, 2020

Dr. Mária A. Deli, M.D., Ph.D.

Associate Editor

Plos one

Dear editor,

Thank you for your careful review of our manuscript, and especially for the constructive criticism of the reviewers. We are also grateful for the opportunity to revise the manuscript, and respond to their critiques. We believe that we have addressed all concerns and, in doing so, have significantly improved the manuscript. We understand that our manuscript is not perfect, but, has important messages for your readers. A point-by-point response to items raised by the reviewers follows.

Response to Editors.

Response to Reviewer

Reviewer #1: Calcagno et al. provided R1 in a timely fashion. Although several major issues

have been addressed, I still do not recommend the publication in this form.

This group has high quality and well written papers, such as PMID 32350353,

which shows an example for this paper's aim as for quality. Although PlosOne

is a lower impact factor journal than the PMID 323250353, PlosOne is Q1 in its

field, therefore high quality of writing and data presentation is expected.

Introduction and Discussion were modified and amended as requested. Still some references are missing, which for example work with Transwell granuloma models, such as PMID 32716613. References 28 and 29 (previously 12 and 13) are still the same, were not corrected, please correct.

Reply: Thanks for suggesting much improved references, they helped to provide more context for our introduction and discussion. A new sentence was added to the discussion as “Walter et al. recently developed an in vitro model to study tuberculous mycobacterial granuloma in central nervous system”. We do apologize for the duplication in reference number 29. It has since been removed from the manuscript. 

Clarification about the repeat numbers was added, although in the text it is still confusing how many repeats were performed. The Figure legend makes

it more understandable.

Reply: A new sentence was added to methods section as “We designed the experiment with 3 replicates, each replicate has 10 samples.” This sentence appears under the subcategory Cytokine measurements. We also added clarification to the legend of the ELISA results. We hope this adjustment makes the text more understandable. 

My two most major concerns are the results section and presentation of the data.

Some figures are missing compared to the previous version and the it is not clear

why or what was corrected. Still the description of the data is very redundant,

although a bit improved compared to the previous version.

Reply: 

Thanks for sharing your concerns. 

The image about TEER test was removed from the manuscript and added as supplementary figure per the reviewer #2. Reviewer #2 brought up a good point that quality control features of our model (ie TEER testing) were not the central focus of this paper, and belonged in the supplementary data. The data on fibroblast growth to prove functionality of the model was also completely removed as per advice from reviewer #2 to avoid redundancy in results. Instead, cytokine growth measurement is used as our surrogate marker for functionality. 

Taking your advice into consideration; we worked to make the results section less redundant. 

Results section changes are as follows 

1) Paragraph 1: Deletion of sentence two to avoid redundancy. The structure of the granuloma is detailed in the figure legend. 

2) Paragraph 2: No changes. The purpose of this paragraph was to outline the progression of our model formation. Development of the macrodevice was the next essential step. 

3) Paragraph 3: No changes. This paragraph outlines the development of the bilayered lung model with its various components. 

4) Paragraph 4: Sentence 2 deleted to avoid redundancy. Sentence 2 was essentially a descriptor of how the BOSGM appears; we agree this was redundant in the setting of the descriptor of appearance in legend of figure 4. We added in a sentence to clarify what the BOSGM is comprised of. We hope this will make it easier for the reader to understand the full picture. 

5) Paragraph 5: We clarified the purpose of cytokine testing. Cytokine expression to show the functionality of the model. 

Overall, we explain our results in an improved and structured fashion � Granuloma formation, macrodevice development � Bilayer lung model development � BOSGM final product � Cytokine test against control. Figure legends do a nice job of describing the visuals without in text redundancy. 

Also the answers for the reviewers' questions were extremely redundant and

minimalistic, which is very unprecedented and unacceptable. Several questions

of the previous revisions were not answered or are not found well in the text.

Marked-up copy should have been provided to help the work of the reviewers.

Reply: We hope that in the round of edits, we provide a more complete view of the changes made and provide more adequate responses. We agree that highlighted version would help you to find extensive changes in the manuscript. We added highlighted Revised 1 for your review. We hope we have adequately addressed any pending concerns in this round of edits. 

Thank you again for taking the time to help us publish our message and share our research with readers of PlosOne. 

All the best,

---

## [Decision Letter · Decision Letter 2]

5 Jan 2021

PONE-D-20-30175R2

Novel three-dimensional biochip pulmonary sarcoidosis model

PLOS ONE

Dear Dr. Mirsaeidi,

Thank you for submitting your manuscript to PLOS ONE. After careful consideration, we feel that it has merit but does not fully meet PLOS ONE’s publication criteria as it currently stands. Therefore, we invite you to submit a revised version of the manuscript that addresses the points raised during the review process.

There are two minor issues left to be addressed as requested by the Reviewer.

We look forward to receiving your revised manuscript.

Kind regards,

Mária A. Deli, M.D., Ph.D.

Academic Editor

PLOS ONE

Reviewers' comments:

Reviewer's Responses to Questions

**Comments to the Author**

1. If the authors have adequately addressed your comments raised in a previous round of review and you feel that this manuscript is now acceptable for publication, you may indicate that here to bypass the “Comments to the Author” section, enter your conflict of interest statement in the “Confidential to Editor” section, and submit your "Accept" recommendation.

Reviewer #1: All comments have been addressed

2. Is the manuscript technically sound, and do the data support the conclusions?

Reviewer #1: Yes

3. Has the statistical analysis been performed appropriately and rigorously? 

Reviewer #1: Yes

4. Have the authors made all data underlying the findings in their manuscript fully available?

Reviewer #1: Yes

5. Is the manuscript presented in an intelligible fashion and written in standard English?

Reviewer #1: Yes

6. Review Comments to the Author

Reviewer #1: Thank you for addressing my comments. Two minor comments are left, and then

I suggest the paper for publication:

For Fig 1. Please insert back the sentence detailing the cell types either in

the figure legends or to the text.

Previously text contained "The presence of multi-nucleated giant cells,

lymphocytes, and macrophages which are aggregated together formed a large

structured granuloma", which is now missing. This information is important,

therefore I suggest its addition to the paper.

Please put a short decription of the model to the Figure 2 legend - explaining

the setup in a compact form.

After these small modifications the manuscript is ready for acceptance.

7. PLOS authors have the option to publish the peer review history of their article (what does this mean?). If published, this will include your full peer review and any attached files.

Reviewer #1: No

---

## [Author Response · Author response to Decision Letter 2]

5 Jan 2021

January 5, 2021, 

Dear editor,

Thank you for your careful review of our manuscript, and especially for the constructive criticism of the reviewers. We are also grateful for the opportunity to revise the manuscript, and respond to their critiques. We believe that we have addressed all remaining concerns and, in doing so, have significantly improved the manuscript. 

Reviewer's Responses to Questions

Comments to the Author

1. If the authors have adequately addressed your comments raised in a previous round of review and you feel that this manuscript is now acceptable for publication, you may indicate that here to bypass the “Comments to the Author” section, enter your conflict of interest statement in the “Confidential to Editor” section, and submit your "Accept" recommendation.

Reviewer #1: All comments have been addressed

2. Is the manuscript technically sound, and do the data support the conclusions?

Reviewer #1: Yes

3. Has the statistical analysis been performed appropriately and rigorously?

Reviewer #1: Yes

4. Have the authors made all data underlying the findings in their manuscript fully available?

Reviewer #1: Yes

5. Is the manuscript presented in an intelligible fashion and written in standard English?

Reviewer #1: Yes

6. Review Comments to the Author

Reviewer #1: Thank you for addressing my comments. Two minor comments are left, and then

I suggest the paper for publication:

For Fig 1. Please insert back the sentence detailing the cell types either in

the figure legends or to the text.

Previously text contained "The presence of multi-nucleated giant cells,

lymphocytes, and macrophages which are aggregated together formed a large

structured granuloma", which is now missing. This information is important,

therefore I suggest its addition to the paper.

Previous text containing the above description was added back to the results section (line 154-156). 

Please put a short decription of the model to the Figure 2 legend - explaining

the setup in a compact form.

A short description explaining the setup of the model in its compact form was added to the Figure 2 legend. 

After these small modifications the manuscript is ready for acceptance.

7. PLOS authors have the option to publish the peer review history of their article (what does this mean?). If published, this will include your full peer review and any attached files.

Do you want your identity to be public for this peer review? For information about this choice, including consent withdrawal, please see our Privacy Policy.

Reviewer #1: No

The above modifications suggested in red have been made to our manuscript. Thank you again for taking the time to help us publish our message and share our research with readers of PlosOne. 

Have a happy new year!

Mehdi Mirsaeidi, MD, MPH

Division of Pulmonary, Critical Care, 

Sleep and Allergy

Director of Sarcoidosis Program

Miami VA Medical Center

Department of Medicine 

University of Miami, Miller School of Medicine

1600 NW 10th Ave # 7072A

Miami, FL 33136

(305) 243-9227

Email: msm249@miami.edu

---

## [Decision Letter · Decision Letter 3]

8 Jan 2021

Novel three-dimensional biochip pulmonary sarcoidosis model

PONE-D-20-30175R3

Dear Dr. Mirsaeidi,

We’re pleased to inform you that your manuscript has been judged scientifically suitable for publication and will be formally accepted for publication once it meets all outstanding technical requirements.

Kind regards,

Mária A. Deli, M.D., Ph.D.

Academic Editor

PLOS ONE

Additional Editor Comments (optional):

Reviewers' comments:

Reviewer's Responses to Questions

**Comments to the Author**

1. If the authors have adequately addressed your comments raised in a previous round of review and you feel that this manuscript is now acceptable for publication, you may indicate that here to bypass the “Comments to the Author” section, enter your conflict of interest statement in the “Confidential to Editor” section, and submit your "Accept" recommendation.

Reviewer #1: All comments have been addressed

2. Is the manuscript technically sound, and do the data support the conclusions?

Reviewer #1: Yes

3. Has the statistical analysis been performed appropriately and rigorously? 

Reviewer #1: Yes

4. Have the authors made all data underlying the findings in their manuscript fully available?

Reviewer #1: Yes

5. Is the manuscript presented in an intelligible fashion and written in standard English?

Reviewer #1: Yes

6. Review Comments to the Author

Reviewer #1: The authors have addressed all my comments. Now the paper is ready for acceptance.

7. PLOS authors have the option to publish the peer review history of their article (what does this mean?). If published, this will include your full peer review and any attached files.

Reviewer #1: No

---

## [Editor Report · Acceptance letter]

27 Jan 2021

PONE-D-20-30175R3 

Novel three-dimensional biochip pulmonary sarcoidosis model 

Dear Dr. Mirsaeidi:

I'm pleased to inform you that your manuscript has been deemed suitable for publication in PLOS ONE. Congratulations! Your manuscript is now with our production department. 

Kind regards, 

on behalf of

Dr. Mária A. Deli 

Academic Editor

PLOS ONE